# Stimuli-Responsive Nanotherapeutics for Treatment and Diagnosis of Stroke

**DOI:** 10.3390/pharmaceutics15041036

**Published:** 2023-03-23

**Authors:** Manisha Choudhary, Sayali Chaudhari, Tanisha Gupta, Dnyaneshwar Kalyane, Bhagwat Sirsat, Umesh Kathar, Pinaki Sengupta, Rakesh K. Tekade

**Affiliations:** National Institute of Pharmaceutical Education and Research (NIPER), Ahmedabad, Department of Pharmaceuticals, Ministry of Chemicals and Fertilizers, Opposite Air Force Station, Palaj, Gandhinagar 382355, Gujarat, India

**Keywords:** stroke, nanotherapeutics, oxidative stress, ischemic stroke, reactive oxygen species, thrombosis

## Abstract

Stroke is the second most common medical emergency and constitutes a significant cause of global morbidity. The conventional stroke treatment strategies, including thrombolysis, antiplatelet therapy, endovascular thrombectomy, neuroprotection, neurogenesis, reducing neuroinflammation, oxidative stress, excitotoxicity, hemostatic treatment, do not provide efficient relief to the patients due to lack of appropriate delivery systems, large doses, systemic toxicity. In this context, guiding the nanoparticles toward the ischemic tissues by making them stimuli-responsive can be a turning point in managing stroke. Hence, in this review, we first outline the basics of stroke, including its pathophysiology, factors affecting its development, current treatment therapies, and their limitations. Further, we have discussed stimuli-responsive nanotherapeutics used for diagnosing and treating stroke with challenges ahead for the safe use of nanotherapeutics.

## 1. Introduction

A stroke is a chronic state associated with inadequate blood supply to the brain tissues due to blockage in neuronal blood vessels [1]. Stroke may also arise due to swelling and neuronal damage caused by the rupture of damaged blood vessels in the brain. Due to this, the blood and oxygen cannot reach the brain tissue. Globally, stroke is the leading cause of mortalities and disability [2,3].

According to the Centers for Disease Control and Prevention (CDC), stroke is the fifth-leading cause of death in the United States. According to recent data, more than 794,000 U.S. people are affected each year due to stroke. Ischemic and hemorrhagic strokes affect as many as 13.1 million people worldwide every year and are the second leading cause of mortalities, with 5.6 million deaths yearly [4,5]. Predictable data concludes that 1 in 4 adults get affected due to a stroke in their lifetime and there are more than 80 million survivors of stroke across the world. The frequency of the disease and commonness of ischemic stroke has been evolved. In 2016, the worldwide occurrences of ischemic stroke reached 9.6 million [6]. Probable motives for the altering occurrence comprise drops in stroke mortality, enhanced secondary preclusion, and improved recognition of stroke [7].

The causes of stroke include atherosclerosis, small vessel disease, arterial dissection, cerebral vasculitis, and reversible cerebral vasoconstriction syndrome. Atherosclerosis is an embolus in the cerebral vasculature initiated by an ulcerated and typically stenotic atherosclerotic plaque in the aortic arch, neck, or intracranial vessels. A patient suffering from atherosclerosis forms thrombi that can block the atherosclerotic vessel [8]. Another cause of stroke is small vessel disease which affects the smaller arteries and arterioles of the brain that can manifest in ways such as lacunar stroke, cerebral microbleeds, and intracerebral hemorrhage [9]. Moreover, arterial dissection in which the intimal layer of an artery develops intramural thrombus can also lead to stroke, specifically in younger patients. Most dissections or tears in arteries that are the source of ischemic stroke are in the extracranial carotid and vertebral arteries and can block the artery at the location of dissection or cause thrombi formation [2]. In vasculitis, vessel wall inflammation leads to luminal narrowing and thromboembolism, causing an ischemic stroke or sometimes even intracerebral hemorrhage. Reversible cerebral vasoconstriction syndrome is present with recurrent sudden onset of headaches and can also lead to ischemic stroke, intracerebral hemorrhage, or focal subarachnoid hemorrhage through vascular spasm and dysregulation [10] (Figure 1).

At the cellular level, when blood supply to a brain region stops, glutamate is released. Glutamate is an excitatory neurotransmitter that releases calcium into the extracellular space. This process facilitates more influx of calcium into the cell that stimulates NMDA receptors. Concurrently, sodium and chloride also enter the neurons, and water flows inside, resulting in cytotoxic edema. This accounts for damage to neurons and glial cells in stroke. Another cause of stroke is mitochondrial dysfunction. Mitochondria are the membrane-bound organelle that plays a role in cellular energy production and calcium homeostasis and produce reactive oxygen species (ROS) [11,12]. Any damage in the mitochondrial membrane directs ROS due to oxidative phosphorylation. The mitochondrial membrane becomes leaky through the formation of mitochondrial permeability transition pores in the membrane; this results in mitochondrial swelling and the accumulation of calcium and ROS [13]. Increased calcium in the neurons exacerbates the level of protease, lipase, endonuclease, and nitric oxide (NO), leading to the breakdown of protein, lipid/triglyceride, nucleotide chain, antioxidant, respectively, and finally, apoptotic cell death [14].

In the current review, we describe and elaborate on stroke, its pathophysiology, factors associated with the development of stroke, current therapies, and their limitations. We have also discussed the emerging opportunities for novel therapeutics such as stimuli-responsive nanotherapeutics for diagnosis and treatment of stroke involving approaches like oxidative stress-responsive nanotherapeutics, thrombosis-responsive nanotherapeutics and pH-responsive nanotherapeutics, and lastly challenges for nanotherapeutics in stroke.

## 2. Factors Associated with the Development of Stroke

Stroke is the second most reason for death and disability worldwide. According to the global burden of disease study, men show a higher incidence of stroke (134 cases per 1 lakh person-year) than women (100 cases per 1 lakh person-year) [15]. This is particularly due to the effect of estrogen and testosterone that modulates the endothelium and vascular system. Moreover, hormone replacement therapy, oral contraceptives, and pregnancy are key factors that increase the risk of stroke in women [16].

The risk of stroke generally increases with age, but people younger than 65 years are more prone to the development of stroke [17,18] (Figure 2). Certain studies show that age influences initial stroke severity and Activities of Daily Living recovery. However, it does not affect neurological recovery, indicating a poor compensatory ability in stroke patients with advanced age [19]. Stroke is also associated with mutations in genetic factors such as cyclooxygenase-2, thromboxane-A synthase 1, integrin subunit β3, and prostaglandin synthase by various mechanisms like a mutation in collagen, cerebral autosomal recessive arteriopathy with infarct. Stroke is also experienced by patients with X-linked Fabry disease and sickle cell disease [20].

Other factors like hyperlipidemia, where a high level of cholesterol in the blood, causes fatty acid deposition in the blood, build plaque in arteries, and make them narrow and firm. This decreases the blood supply to the brain [21,22]. A similar situation arises in hypertension. Hypertension leads to increased stress on the endothelium, and the resulting degradation of endothelium alters the interactions between blood cells and endothelium, leading to the formation of thrombi that blocks blood flow to the brain [23]. Plaque formation is also seen in diabetes mellitus. The lack of insulin in diabetes increases blood sugar levels. It gradually builds up a clot or fat deposit within the vessel that supplies blood from the periphery to the brain. This may cause blockage of blood vessels leading to a reduced blood supply to the brain, finally leading to stroke [24].

Stroke may also be developed by oxidative stress [25]. Overproduction of ROS involving oxygen, hydroxyl radical (˙OH), hydrogen peroxide (H_2_O_2_), and NO increases the release of the cortisol stress hormone, creating massive pressure in the brain. A study suggests that mental stress may increase the signal from the amygdala to bone marrow, increasing the production of white blood cells. These can build up in arteries leading to the clogging of arteries and the development of stroke [26].

## 3. Current Treatment Strategies Used to Combat Stroke

The treatment modalities of stroke are governed by the type of stroke: ischemic stroke or hemorrhagic stroke. The treatment regimen particularly focuses on the reperfusion strategies engaged in arterial recanalization, prevention and treatment of complications associated with the stroke, intonation or inhibition of inflammatory response, and neuroprotective strategies engaged in a metabolic and cellular target. However, having the multi-mechanistic pathophysiology of stroke, a multistep approach for the treatment of stroke is recommended [27,28,29].

In this section, we have discussed the current status of the treatment regimen available for stroke. In addition, a detailed portrayal of the pharmacological agents is discussed here to understand the above-mentioned strategies for treating stroke and the mechanism rationale behind the treatment used (Table 1). 

### 3.1. Ischemic Stroke

In an ischemic stroke, the arteries in the brain become narrow, leading to the formation of blood lumps and disturbance in blood flow, further progressing in blockage of arteries. The vast proportion of strokes are ischemic, brought on by an occlusion in an artery [62]. This can also be triggered by fragments of plaque due to atherosclerosis. Ischemic stroke is further categorized as thrombotic and embolic stroke [63]. A thrombotic stroke occurs when a blood clot (thrombus) develops in one of the arteries delivering blood to the brain, thereby hindering the blood flow, whereas in embolic stroke, the thrombus or other debris develops elsewhere and enters the brain followed by blockage of blood supply [64].

One of the critical strategies to combat ischemic stroke is thrombolysis. Clinically, it is performed by the recombinant tissue plasminogen activator (rtPA) that catalyzes the cleavage of zymogen plasminogen to form active plasmin and converts the insoluble fibrin clot into the soluble product by deprivation of interlinked fibrin monomers [30]. Consequently, this clears the clot and aids the reperfusion of the ischemic tissue. Another approach is antiplatelet therapy using aspirin or NSAIDs (Non-Steroidal Anti-Inflammatory Drugs). These drugs, in low doses, break down the platelet thrombus, clearing the blood vessels to maintain smooth blood circulation [2,38]. Clopidogrel or aspirin-dipyridamole are substitutes for aspirin in secondary stroke as they are more effective than the latter [40].

Simultaneously, Carloni et al. and Shehadah et al. showed that the administration of statins may reduce cholesterol deposited in blood vessels of the rodent models of ischemic stroke, attributing to the neuroprotective properties against neuronal damage and infarct dimensions [65,66]. Similarly, the agents such as filgrastim and leucostim, which are genetically engineered recombinant human Granulocyte colony-stimulating factors (rhG-CSF), showed neuroprotective, neuro-operative, and anti-inflammatory effects in ischemic stroke [42,43]. Agents like Edavarone inhibit lipid peroxidation and lipoxygenase pathways accounting for neuroprotection from ischemia or reperfusion-induced vascular endothelial cell injury. It also has a role in brain edema, neurological deficits, and delayed neuronal death [44]. Moreover, caffeinol, a combination of caffeine and ethanol, was checked for its neuronal activity by Belayev and co-workers. The administration of caffeinol post-reperfusion showed a substantial decline in volumes of cortical infarction as well as a rise in neurological scores [49].

Ischemic stroke can also be managed by endovascular thrombectomy. This refers to removing the thrombus surgically and clearing the way for blood flow to the respective ischemic tissue [53]. The primary guideline used for the treatment of stroke by endovascular thrombectomy recommends that the treatment should start within 6 h to 24 h of the inception of the stroke [2,67]. Altogether, ischemic stroke is treated by increasing the reperfusion via thrombolysis, antiplatelet therapy, endovascular thrombectomy, neuroprotection, neurogenesis, reducing neuroinflammation, oxidative stress, excitotoxicity [68,69]. As there are delays in reaching patients to the hospitals, it seems very challenging and difficult to treat patients with ischemic stroke [70].

### 3.2. Hemorrhagic Stroke

Hemorrhagic stroke, also known as intracranial hemorrhage (ICH), arises when a weak blood vessel breaks and bleeds into brain tissue. The risk of ICH rises where hypertension is not treated [71]. The leaked blood exerts pressure on the brain and causes swelling and neuronal damage. Intracerebral and subarachnoid are the two types of hemorrhagic stroke. The most common type of hemorrhagic stroke is intracerebral hemorrhagic stroke, which leads to bleeding inside the brain. In contrast, subarachnoid hemorrhagic stroke is not as common as it causes bleeding in the area surrounding the brain and the tissues that cover the subarachnoid part [72,73]. Unlike an ischemic stroke, less progress has occurred in treating hemorrhagic stroke [74].

Hemorrhagic stroke may arise due to chronic hypertension; therefore, intense antihypertensives are preferred to lower the risk of stroke [56]. Moreover, it is also controlled by hemostatic therapy, which includes agents that enhance coagulation and stop the bleeding, like an activated form of natural factor VII, i.e., recombinant factor VIIa [59]. This therapy is most beneficial in the warfarin-related ICH in patients with smaller hemorrhages and provides more promising baseline examination results. However, neuronal bleeding can result in the formation of large clots in the brain parenchyma, which may prove fatal. These are removed by surgical evacuation so that the thrombin and iron are ultimately removed from neurons and glial cells along with the clot. These both are lethal compounds triggering edema and leading to inferior impairment of brain tissue [74]. Hence, collectively hemorrhagic stroke is managed by lowering the acute elevated blood pressure, hemostatic therapy, surgical evacuation, discontinuation of warfarin, and correcting the INR (International Normalized Ratio) in the combination of vitamin K (rapid IV or parenteral administration) [74].

## 4. Limitations of Current Therapies

The current treatment for stroke cannot meet the desire to treat the disease effectively. The reasons include: (i) Well-established treatments are not accessible to every patient due to a lack of better methods to deliver these therapies to patients, (ii) The disadvantages associated with the treatment regimen of current therapies, (iii) Unavailability of better monitoring for negative consequences allied with current therapies. In addition, although novel oral anticoagulant therapies promise to improve the condition of patients by preventing/treating stroke, these formulations should be enhanced to inverse their hemorrhagic complications, too [75].

It is also important to note that most patients with stroke are not suitable for the treatment with thrombolysis, and about 5% of them receive rtPA. The IV administration of rtPA is limited to an inadequate time space as well as a modest success rate. Not only this, but also the maximum number of patients visiting the stroke care units either cannot use rtPA treatment, such as recent surgery or a bleeding diathesis, or they are not given rtPA therapy in the stipulated time, which is assigned for the thrombolytic therapy [27,28,29].

The neuroprotective drugs show an intimidating advantage in the phase of animal study research. However, it has been incapable of reproducing the same effect in human clinical trials. The limitations of this include permanent disability due to treating stroke in its acute phases and preventing sequels [76].

## 5. Stimuli-Responsive Nanotherapeutics for Diagnosis and Treatment of Stroke

The concept of employing nanoparticles as a carrier system for special purposes, drug and active agent delivery has achieved great development. The objective of designing these systems, which generally consists of drug, carrier, coating and targeting moiety, is the controlled drug release, the maintenance of drug concentration within the therapeutic window for an adequate time period, and drug delivery to the target tissue [77]. Among the several types of nanoparticles, carbon nanoparticles like conical nanocarbon, quantum dots, and carbon nanodots. have a few advantages, which include being less expensive, more accessible, safe to handle, and available in a variety of formulations [78,79]. The environment of the ischemic tissue is different from that of normal cerebrovascular tissue. This is due to the alteration in blood flow, deprived oxygen levels, and upregulation of platelets. These distinct features can be used to target the nanotherapeutics toward the damaged tissue [80]. The following section highlights the abnormal environment of the neuronal tissue bearing stroke and how these environmental conditions can be used for delivering stimuli-responsive nanotherapeutics (Table 2). 

### 5.1. Oxidative Stress-Responsive Nanotherapeutics

Oxidative stress is defined as an imbalance between the production of ROS/free radicals and the antioxidant agents, leading to cell and tissue damage. Oxidative stress is due to rising levels of ROS, including hydroxyl radical, superoxide anion (O_2_^−^), NO, and peroxynitrite (ONOO^−^), which are usually generated after reperfusion injury. ROS are unstable and possess high reactivity causing lipid peroxidation, protein denaturation, cell organelles, and nucleic acids damage, and finally, activating the pathways like necrosis, autophagy, or apoptosis during an ischemic stroke. This, in turn, increases the cerebral infarct size, damages the cerebral vasculature and neuronal network, and disturbs the blood–brain barrier, which causes brain edema and hemorrhage [101,102].

ROS leads to oxidative DNA damage that can persist from several days to six months after an incident of stroke. Damage to DNA can progress in the demyelination of neurons and produce injuries to the axon. Hence, they can be used as a therapeutic target to reverse neuronal damage and prevent further damage to cerebral tissue. However, an increase in ROS levels activates the antioxidant defense system that helps neutralize the ROS. These antioxidants include enzymes like Superoxide Dismutase (SOD), Catalase (CAT), Glutathione Peroxidase (GSH-Px), and Glutathione-S-Transferase (GST). When the levels of ROS exceed that of antioxidants, there is oxidative stress, and thus tissue damage occurs [25,103,104,105]. Hence, using therapeutic agents acting as antioxidants becomes an ideal approach to managing stroke/ischemic reperfusion injury. However, this approach is limited due to poor blood–brain barrier (BBB) penetration, short circulation half-lives, and poor site-specific brain delivery of antioxidants. To combat these issues, nanotechnology is being employed as a treatment strategy where ROS can be used as markers for the activation of nanosystems to release the therapeutic agent. Owing to their large surface area to volume ratio, nanotherapeutics show higher active electrons on the outer surface, indicating good catalytic and antioxidant activity. Thus, researchers aim to develop ROS-responsive nanotherapeutics for stroke and stroke-associated ischemia-reperfusion injury [101].

In this regard, Lv et al. developed a nanosystem encapsulated in a shell of an RBC membrane with a stroke-homing peptide (SHp-RBC) [81]. The core consisted of a neuroprotective agent NR2B9C encapsulated into a Boronic Esters Modified-dextran (PHB-Dextran) polymer coat. Together they form SHp-RBC-NP/NR2B9C nanoparticles (NPs). RBC membrane prolongs the circulation time up to 48 h, SHp acts as a target for the ischemic stroke neurons, and PHB-Dextran acts as an H_2_O_2_-responsive polymer. Once inside the neuron, the RBC membrane and PHB-dextran were degraded due to H_2_O_2_, releasing the NR2B9C into the neurons.

Furthermore, NR2B9C binds to postsynaptic density protein and hinders its interactions with the NMDA receptor, thus inhibiting the excess production of NO, a toxic agent. In vivo studies on middle cerebral artery ischemia (MCAO) with Sprague-Dawley rat models were performed. The neuroscore assessment was conducted to evaluate the anti-ischemic-stroke efficacy of the SHp-RBC-NP/NR2B9C. The SHp-RBC-NP/NR2B9C-treated groups showed less neurological damage, and reduced brain infarct area than the group only treated with the NR2B9C agent, suggesting the principle of ROS-responsive drug delivery for stroke (Figure 3) [81].

One of the polymers explored as ROS-responsive agents in stroke therapy is Poly(Propylene Sulphide) (PPS). It contains Sulphur (II) atoms that, in contact with ROS, change their polarity and are converted to their more soluble/polar form. Solubilization of this polymer leads to the release of drugs incorporated within it. This phenomenon is explored by researchers to formulate a ROS-responsive drug delivery system for the treatment of stroke. Rajkovic et al. prepared PPS NPs coated with polyethylene glycol to enhance the circulation time of the NPs. After injecting these PPS-NPs in the MCAO rat model, the brain infarction volume was significantly reduced, and the damage to BBB was also decreased. In all, 69% of the neuronal loss post-stroke was reduced to 41% after 24 h of PPS-NPs injection [82].

Researchers developed copolyoxalate-incorporated vanillyl alcohol (PVAX) NPs. Copolyoxalate contains peroxalate ester linkages in its backbone, which are ROS-responsive. Mainly, when copolyoxalate polymer comes in contact with H_2_O_2_ and water, there is hydrolytic degradation of the polymer due to peroxalate linkages. This property of copolyoxalate was used to develop PVAX. Here, VA is vanillyl alcohol, a plant-derived antioxidant for ischemic brain injury. On reaching the ischemic tissue, high levels of H_2_O_2_ induced the breakdown of PVAX NPs, releasing the VA. When tried on a mouse ischemic reperfusion injury model, there was a reduction in the levels of inflammatory markers like TNF-α and IL-1β, suggesting the anti-inflammatory activity of the PVAX NPs in ischemic stroke [83].

Apart from therapeutics, ROS responsiveness is also being considered for the diagnosis of stroke. Electric pulses, the agents that can react with ROS to yield measurable signals in the form of luminescence, can be used as diagnostic aids in pathological conditions associated with abnormal ROS levels like stroke and reperfusion injury. This was considered by Lee and coworkers, who engineered hydroxylenzyl alcohol-incorporating copolyoxalate (HPOX) NPs to bioimage the pathological conditions involving high H_2_O_2_ concentrations like stroke and other neurovascular diseases. HPOX NPs were fabricated and loaded with Rubrene which acts as a fluorophore. At high H_2_O_2_ concentrations, a chemiluminescent reaction occurs between the peroxalate ester linkages in HPOX and H_2_O_2_ to yield dioxetanedione. Dioxetanedione is a high-energy intermediate that excites rubrene to liberate strong emissions at 565 nm. This emission is recorded as a measure of H_2_O_2_ levels. With a linear increase in the concentration of H_2_O_2_, the emission by rubrene was also increased, suggesting that rubrene-loaded HPOX NPs can be used to diagnose diseases/disorders with abnormal H_2_O_2_ concentration with good specificity. In combination with this, the HPOX NPs were also evaluated for their antioxidant activity. Thus, the HPOX NPs show great potential as a diagnostic and therapeutic aid in H_2_O_2_-associated pathological conditions [84]. Similar bioimaging ability was also reported by PVAX NPs [83].

Another group of scientists fabricated ROS-responsive nanoprobes to quantify ROS in stroke. They developed lanthanide (Ln) doped NPs consisting of Ln as a luminescence agent, two amphiphilic polymers, namely, 1,2-Distearoyl-sn-glycero-3-phosphoethanolamine-Poly(ethylene glycol) (DSPE-PEG_2000_) to impart aqueous solubility to the NPs and DSPE-PEG_2000_-VHPKQHR to target the vascular cell adhesion protein that is overexpressed in stroke ischemic tissues. The NPs were decorated with a ROS-responsive photosensitizer IR-783 dye. After reaching the ischemic tissue, the ROS, specifically hypochlorous acid (HOCl) and ˙OH oxidize the IR-783 on the NPs and inhibits its photosensitization reaction with Ln, thereby decreasing the luminescence intensity. The IR-LnNPs can thus be used as a probe to detect the ROS levels that are a hallmark of stroke onset and a post-stroke symptom [85].

### 5.2. Thrombosis Responsive Nanotherapeutics

Thrombus formation is associated with ischemic stroke and sometimes hemorrhagic stroke. Hence, molding the nanotherapeutics and making them thrombus-responsive can be a treatment strategy for managing stroke. The mechanism for the formation of a platelet thrombus is as follows:

The blood vessel wall contains an inner lining of the endothelial cell, which maintains the vasculature. The endothelial lining contains three tissue factors: NO, prostacyclin, and ectonucleotidases CD39, which prevents the formation of thrombus and also has collagen that imparts elasticity and strength to the vessels. Altogether, these are responsible for the maintenance of the closed circulatory system. Any disruption in the vessel wall or endothelial cell exposes the tissue factors and collagen to the flowing blood. Collagen interacts with platelets and activates a series of pathways resulting in the accumulation and activation of platelets. In contrast, the tissue factors initiate the generation of thrombin, which converts the fibrinogen to fibrin and activates the platelets. The activated platelets and fibrin form a platelet thrombus that interrupts blood flow to the brain, leading to stroke [106] (Figure 4).

Since thrombus formation remains a hallmark of stroke, it can be used as a stimulus to trigger the nanotherapeutics to release drug payloads accurately at the site of stroke injury.

Considering this, Lu et al. focused on fibrin in the thrombus as a targeting agent and fabricated rapamycin (RAPA)-loaded fibrin targeting micelles designated as CPLB/RAPA. Here, C represents the CREKA peptide, which is a fibrin binding-peptide, and PLB is PEGylated LysineB—together, the CPLB is a self-assembling polymer with ROS-scavenging properties. RAPA is an mTOR signaling pathway inhibitor that prolongs the survival of neurons by initiating the cleaning of cell debris and reducing inflammation. When CPLB/RAPA is combined with BODIPY (Borondipyrromethene-a NIR probe) for tracing the concentration of the micelles, the regions with fibrin clots having damaged vasculature showed higher fluorescence intensity. This relates that the micelles were predominantly accumulated in the region of fibrin clots due to the presence of fibrin-responsive CREKA peptide. Thus, higher micelle concentration at injured vasculature, as in stroke, results in higher rapamycin release at the site, followed by scavenging the ROS, finally prolonging neuronal survival [87] (Figure 5).

In stroke injury, complex cellular events occur, such as oxidative stress, neuroinflammation, and brain vasculature, leading to the formation of a microthrombus that can cause the death of neurons. Hence, a nanosystem targeting the microthrombus can be helpful in stroke injury. With this view, Manickam and coworkers developed SOD1 *cl*-nanozymes. SOD1 or catalase are antioxidant enzymes that can be used in the treatment of ischemic reperfusion injury in stroke due to their ability to scavenge ROS and prevent further damage to neurons. However, these enzymes cannot be delivered as such due to their physicochemical characteristics and stability issues. Fabricating them as nanozymes solves the problem. Here, the negatively charged SOD1 enzyme binds electrostatically to positively charged cationic block copolymer further, which gets cross-linked to form *cl*-nanozymes. These *cl*-nanozymes were then tested for therapeutic efficacy in the rat MCAO model. The *cl*-nanozymes collectively caused a 59% decrease in the brain infarct volume and a 70% improvement in sensory-motor activities [107]. Fluorescent double staining shows the localization of cl-nanozyme in the blood vessels, and the damaged artery was the only region showing intense fluorescence in the whole brain. *cl*-nanozymes become localized in the fibrin network of the thrombus and exert their effects on the neurovascular unit of the brain [86].

An enzyme highly expressed in ischemic tissue of stroke is thrombin [108]. Researchers have found significantly elevated levels of thrombin at 3 to 8 h post-stroke episodes. Hence, formulating NPs responsive to thrombin can be one of the approaches to managing post-stroke complications, such as the development of blood clots and brain edema [109,110]. In this regard, Guo et al. developed a block copolymer of PEG, poly(ε-caprolactone) (PCL), and a thrombin cleavable peptide-NH_2_-norleucine-TPRSFL-CSH (T). They further used it to fabricate AMD3100-conjugated PEG-PCL-T-PEG NPs loaded with glyburide to target thrombin in the ischemic tissue and facilitate site-directed delivery of glyburide. Glyburide inhibits the Sulfonylurea Receptor 1-Transient Receptor Potential Melastatin-4 found in the brain. It modulates the flow of ions in a manner to reduce the edema, while AMD3100 is a CXCR4 antagonist abundant in ischemic tissue, hence assisting the NPs in reaching the ischemic site. In vivo experiments in the MCAO mouse model show a 5.5 times higher accumulation of these NPs in ischemic tissue compared to PEG-PCL NPs. Moreover, their survival rate was improved, and cerebral infarct volume was reduced by 36%, with significant improvement in the neurological scores [88].

Thrombus responsiveness of nanotherapeutics is also explored for the diagnosis of stroke. In this regard, an EP-2104R was developed. It binds to fibrin in the thrombus and consists of 11 amino acids with the N and C terminals modified with 2 GdDOTA moieties. The EP-2104R shows the binding ability for two sites on the fibrin protein of rats, rabbits, pigs, mice, and dogs. In vivo studies demonstrated that EP-2104R has 100 times more affinity towards fibrin than fibrinogen and a higher molecular relaxation (25 times higher) ability than the GdDOTA alone. All this suggests using EP-2104R as an MRI contrast agent for detecting a blood clot in a stroke [89].

### 5.3. pH-Responsive Nanotherapeutics

The brain has an intracellular pH of 7.2, maintained by the active and passive transport of ions. During a stroke, this pH is hampered by the decreased outward movement of CO_2_ from the cells due to poor perfusion, resulting in the accumulation of CO_2_ inside the cells. With 3–4 folds more CO_2_ and less oxygen than normal, the cell performs anaerobic glycolysis to convert glucose into lactate, causing up to three times increase in lactate production by glial cells and neurons. This results in the generation of H^+^ ions and lactic acidosis. Another mechanism of acidosis takes place after an ischemic episode. Here, the depolarization waves spread to the cerebral regions and produce a depression known as spreading depression (SD). It increases the energy demand, but due to less oxygen, anaerobic glycolysis is preferred for energy generation resulting in high lactate production. Hence, the pH of ischemic penumbra further decreases up to 6.5, resulting in moderate acidosis. Acidosis is also coupled with the formation of free radicles after an ischemic stroke. It increases the production of H^+^ ions and decreases the pH in the cells. When this pH is less than 6.3–6.4, cellular damage is activated in ischemic tissue [111,112,113,114]. Hence, the acidic pH is a hallmark of stroke and can be used to design pH-responsive nanotherapeutics that could serve in the treatment as well as diagnosis of stroke.

The pH-responsive polymers are being explored for fabricating NPs to manage stroke. One such polymer is chitosan. Chitosan acts as a pH-responsive polymer and gets degraded at the acidic pH of the cerebral ischemic penumbra. Considering this, Tóth et al. fabricated nimodipine pH-responsive chitosan NPs to manage stroke. They used nimodipine, as it is an L-type voltage-gated Ca^2+^ channel antagonist that relieves the constricted vessels that help prevent stroke due to cerebral ischemia and also limits SD. The pH triggered the delivery of chitosan-nimodipine NPs, thus targeting the acidic pH (pH 6.9–7.1) in ischemic penumbra neurons, thereby decreasing the systemic toxicity of the drug. Since nimodipine is a vessel relaxant, an increase in cerebral blood flow (CBF) will indicate nimodipine release from the NPs. In vivo studies showed that administering the chitosan nimodipine NPs in an animal group with induced cerebral ischemia increased the CBF indicating that the polymer was degraded at acidic pH and nimodipine was released. This was not seen when chitosan-nimodipine NPs were administered without inducing cerebral ischemia, indicating that the chitosan was not degraded at the physiological pH of the cerebrum [90].

Exploring other polymers for pH-responsive drug release in ischemic stroke, He et al. used methoxy poly(ethylene glycol)-block-poly (2-diisopropyl methacrylate) (PEG-PDPA) polymer [115] and synthesized succinobucol (SCB) polymeric NPs (PP) and camouflaged them with 4T1 cell membranes (MPP) to give MPP/SCB to increase the targeting ability of the NPs in the cerebral ischemic tissue. SCB acts as an anti-inflammatory and antioxidant drug that will relieve inflammation and decrease oxidative stress in the ischemic tissue. The pH-responsive behavior of these NPs was evaluated by incubating them at different pH and estimating their structure. As the pH was decreased from 7.4 to 4.7, the structure of the NPs was disturbed from perfectly spherical to barely spherical, suggesting the degradation of polymeric NPs. In addition, they showed a pH-dependent SCB release profile with a significant increase in the release at acidic pH (4.7 and 5.4) than at physiological pH (7.4) [91].

Further, the in vivo biodistribution of the NPs was studied in tMCAO rat models using DiD fluorescent probe. Both the NPs, namely, PP and MPP, were labeled with DiD. The NPs were concentrated and found in significantly higher amounts in the ischemic hemisphere than in the normal hemisphere suggesting the pH responsiveness of the NPs. The infarct volume and the neurological deficit score were significantly reduced in the groups treated with MPP/SCB than those treated with PP/SCB and non-treated groups [91] (Figure 6).

As discussed earlier, ROS causes further damage after an ischemic reperfusion injury. Though the nitroxyl radicals are proven free radicle scavengers, an efficient approach to deliver these is still in research. In this regard, Marushima et al. developed PEG-coated NPs of 4-amino-TEMPO (4-amino-2,2,6,6-tetramethylpiperidine-1-oxyl) coblock polymer. TEMPO serves as a nitroxyl radical donor, while the 4-amino group was added to impart pH responsiveness to the NPs. In vivo MCAO rat models showed that at physiological pH, the NPs remain intact, and on reaching the ischemic tissue, where the pH is below 7.0, the amino groups are protonated. As a result, the NPs collapse, releasing the nitroxyl radical at the site. The cerebral infarct volume was significantly reduced in the animals treated with PEG-4-amino-TEMPO NPs than in those treated with saline. This was supported by a decrease in protein carbonyl levels in the PEG-4-amino-TEMPO NPs treated groups suggesting a decline in the oxidation of proteins brought about by a decline in ROS levels [92].

pH responsiveness of nanotherapeutics is also explored for the diagnosis of stroke and cerebral ischemia. In this view, Gao et al. developed polymeric Fe_3_O_4_ NPs that can be used as Magnetic Resonance Imaging (MRI) agents [94]. At a physiological pH (pH 7.4), the two polymers, namely, methoxy polyethylene glycol (PEG) and poly(β-amino ester) (PAE), self-assemble to form polymeric micelle PEG-PAE. These NPs had Fe_3_O_4_ in their core. PEG increases the circulation half-life of the NPs, while the PAE polymer contains ionizable tert-amino groups and thus acts as a pH-responsive unit. PAE becomes protonated and solubilized at an acidic pH (pH ≤ 6.8), thus releasing the Fe_3_O_4_ at the acidic site. Similar NPs were formulated by replacing PAE with pluronic F127, a non-pH-responsive polymer. Upon administering in vivo, it was observed that the PEG-PAE-Fe_3_O_4_ NPs released Fe_3_O_4_ due to ionization of PAE, and the MRI signal was increased, which was not the case with PEG-F127-Fe_3_O_4_ NPs, where the MRI decreased since F127 was not able to degrade and release Fe_3_O_4_ [93]. Similarly, in another approach, the PAE was replaced by poly(β-amino ester)/(amido amine) (PAEA), and PEG was substituted with methyl-PEG to give methyl-PEG-PAEA-Fe_3_O_4_ NPs. They also found a significant release of Fe_3_O_4_ in acidic ischemic tissue. This concluded that methyl-PEG-PAEA-Fe_3_O_4_ could be used as an MRI agent in diagnosing diseases involving acidic environments, such as ischemic stroke [94].

### 5.4. Photo-Responsive Therapy

Near-infrared light (NIR) was very frequently used to regulate the release of the drug because it is safer and has deeper tissue penetration in comparison to visible and UV light [116]. Low-energy laser irradiation (LELI) triggers a photochemical response in the cell without producing heat by utilizing low-powered laser energy with wavelengths between 600 to 1100 nm; this procedure is known as Photobiomodulation (PBM) [117]. The photothermal effect, upconverting nanoparticle, and photon activation were the three mechanisms behind the responsive delivery system [118]. Transcranial infrared laser therapy (TLT) is a type of laser therapy that can permeate the skull and photostimulated brain regions positioned at least a few centimeters under the skull [119].

The effect of NIR, TLT, and PBM on stroke treatment has been studied broadly. Meyer et al. 2016 utilized TLT on a small rabbit clot embolic stroke model (RSCEM) to analyze the clinical outcome of stroke. At 24 h post-embolization, the rabbits were subjected to behavioral analysis to calculate the effective dose for stroke (ES50) or amount of clot (mg) that causes neurological impairments in half of the group. Three treatments with TLT of 111 mW at 2 h post-embolization for 2 min (6.47 ± 1.06, *n* = 17) were found to be significantly more effective in terms of behavioral outcomes compared to the previously utilized regimen, which is 3.09 ± 0.51, *n* = 15. When given 2 h after embolization, TLT dramatically improves behavioral symptoms [117]. One study examined how much stroke volume improved when phototherapy was administered to newborns with jaundice. This study comprised 29 hospitalized infants who were getting phototherapy. Participants’ echocardiograms were taken before therapy and again after 24 h of treatment. The therapy was associated with a reduction in stroke volume in 19 neonates. The average stroke volumes were 6.99 ± 2.17 and 6.55 ± 1.85 L/m^2^ before and after receiving phototherapy, respectively [120]. To perform brain PBM in a mouse model, researchers created a lightweight, small, and simple platform of miniature electrical devices made up of packed light-emitting diodes (LEDs) that contain a flexible substrate. This device system is used to examine the protective and restorative properties of PBM attached to the bare skulls of mice in a stroke model of middle cerebral artery blockage and photothrombosis. PBM with an LED array of 630 nm significantly enhances spatial memory and learning in the severe poststroke phase, reduces activation of AIM2 inflammasome and pyroptosis mediated with inflammasome alters polarization of microglia in the cortex and hippocampus, and decreases the volume of the infraction and neuron impairment. PBM has the potential to lessen the severity of cognitive impairment following stroke by reducing damage caused by acute ischemia injury [121] (Figure 7).

### 5.5. Enzyme Responsive Therapy

Nanocarriers with molecular targets sensitive to endogenous enzyme overexpression in either the intracellular or extracellular settings are called enzyme-responsive nanocarriers [122]. Since enzymes play a crucial part in a wide variety of biochemical events—their expression being stimulated may be employed to cause the release of medication at strategic sites. Based on the enzymes that were upregulated, many different types of enzyme-responsive nanocarriers were produced. These included polymers, nanogels, liposomes, nano-complex, nanoparticles, and nanovesicles. Enzyme-responsive nanocarriers containing the substrate of the enzyme as the linkage will break upon contact with the enzymes, causing them to swell and release more of the drugs inside [123].

It has been postulated that TAK1 (transforming growth factor beta-activated kinase 1) activation, a major MAP3K upstream of numerous inflammation-regulating cascades, causes microglia/macrophages (Mi/MΦ) to adopt a proinflammatory phenotype and hence exacerbates reperfusion/ischemia brain injury. Middle cerebral artery occlusion (MCAO) and reperfusion were performed on young adult mice for 1 h. Mi/MΦ-specific knockouts induced with tamoxifen or selective inhibitor 5Z-7-Oxozeaenol administration following MCAO were both used to target TAK1. Specifically, targeting TAK1 in Mi/MΦ significantly reduced neurological impairments. TAK1 enhances proinflammatory and harmful Mi/MΦ responses, which contribute to inflammation induced by ischemia/reperfusion, brain damage, and maladaptive behavior. As a result, the inhibition of TAK1 is a prospective treatment to optimize long-term outcomes following stroke [124]. One research project aimed to determine how Sult2b1 and cholesterol sulfate contributed to neuroinflammation following ischemic stroke. Middle cerebral artery blockage was performed temporarily on both Sult2b1−/− and wild-type animals. It was shown that Sult2b1−/− favored the macrophages’ pro-inflammatory polarization by manipulating the levels of reactive oxygen species (ROS), nicotinamide adenine dinucleotide phosphate (NADPH), and the AMP-activated protein kinase (AMPK)—cAMP-responsive element-binding protein (CREB) signaling pathway activation. Cholesterol sulfate, which encourages macrophages to become polarized toward an anti-inflammatory condition via metabolic control, may help to mitigate this trend. This research linked macrophage polarization to the inflammatory and metabolic processes contributing to ischemic stroke [125]. To prevent acute ischemic stroke, the antioxidant enzyme Peroxiredoxin-1 (Prdx1) regulates microglia linked with stroke. Prdx1 prevents microglial cell death in ischemia and regulates the immune response by stimulating the development of a novel microglia cluster, i.e., stroke-associated microglia (SAM), in the ipsilateral hemisphere. Prdx1 regulates cell death pathways in the brain cells and may be a pivotal target protein for mitigating the effects of ischemic stroke. When it comes to preserving the function of microglia and treating I/R injury of the brain, Prdx1-dependent SAM may serve as both a biomarker as well as a therapeutic target [126] (Figure 8).

### 5.6. Ultrasound Responsive Therapy

Directed thrombolysis extensively uses ultrasonic waves, a form of energy that may propagate into bodily tissue and cause numerous distinct physical effects. It has been discovered that the nanomaterial perfluorohexane (PFH) can undergo a phase transition (PT) when subjected to an explosive ultrasound pulse [127]. Ultrasound’s non-ionizing properties, low cost, and lack of invasiveness are just a few of its numerous benefits. Furthermore, based on frequency and exposure, it can penetrate deeply, allowing for site-specific results with minimal side effects [128]. Exogenous signals, such as ultrasound, are utilized to stimulate the release of thrombolysis drugs [123].

Recent advances in ultrafast ultrasonography may provide neuroradiologists with a potent new tool for urgent imaging and long-term monitoring of cerebral perfusion. In this investigation, researchers used a mouse model of thromboembolic stroke to compare a group of untreated mice, the control (N = 10), with a group treated with the current gold standard therapy of pharmacological stroke (N = 9). Research demonstrates that brain perfusion during an ischemic stroke may be tracked using ultrafast ultrasound imaging in mice. Reperfusion monitoring after thrombolytic therapy within the first two hours after the stroke start correlates with ischemic lesion measurements taken 24 h later. This is the first step before bringing new research into clinical settings, and it brings new capabilities to investigate ischemic stroke in preclinical animal models [129].

In addition, scientists have developed a platelet hybrid microglia platform that can be polarized to an anti-inflammatory phenotype using ultrasonic irradiation, allowing for more focused repair of damaged brain tissue following a stroke. The microglia platform, when administered intravenously, exhibited the expected anti-inflammatory polarization at the stroke site under insonation control and expedited the M2-type endogenous microglia polarization for long-term stroke healing. Decreased apoptosis, increased neurogenesis, and improved functional recovery all lead to positive outcomes, underscoring the significance of the microglia platform in stroke treatment [130].

## 6. Challenges for Nanotherapeutics in Stroke

Nanotherapeutics are complex and sensitive in terms of their method of preparation, applications, and stability [80]. Nanotherapeutics such as polymeric nanocarriers, polymeric micelles, liposomes, exosomes, and inorganic NPs are proposed for stoke treatment and diagnosis [131]. In addition, a few NPs are evaluated by scientists for the diagnosis of stroke using MRI, positron emission tomography and computed tomography, and ultrasound [80].

Although nanotherapeutics claim potential use in therapeutics and diagnostics, there are certain challenges in their clinical translation. Owing to their small size, the nanomaterials have a large surface area exposed to biological fluids, making them vulnerable to interactions with biological substances, especially blood components, proteins, and macrophages. Moreover, the reticuloendothelial system (RES) sometimes uptake the NPs thinking they are foreign microorganisms. This leads to the establishment of additional safety data of NPs against interaction with biological agents [132].

Moreover, the bioavailability of the NPs also remains a question since the cerebral vasculature is slightly disrupted after stroke, and the NPs can penetrate at a higher rate in the brain. However, these NPs can leak out of the brain in a similar manner in which they entered. Thus, the retention of NPs at the therapeutic site is a matter of concern. However, several attempts are being made to develop site-targeted nanotherapeutics by targeting ligands or inducing their stimuli responsiveness. This will provide a longer retention time in the brain and avoid peripheral side effects [80].

The nanomaterials often show cytotoxicity; hence, cytotoxicity assessment studies are to be carried out for every type of nanomaterials. However, the choice of in vitro parameters for assessing the cytotoxicity remains a matter of concern, as they must mimic the in vivo conditions. This incurs extra time, studies, and costs for the nanoformulations [80].

Moreover, technology transfer from the lab to the industrial scale is often associated with changes in the particle size, surface charge density, morphology, amount of drug encapsulated, quantity of residual matter and stability. Apart from this, industries appreciate the green synthesis of formulations. Still, NP synthesis often involves organic solvents and toxic compounds, which may generate a large amount of industrial waste and cause environmental issues. Hence, the laboratory-to-industrial translation of nanotherapeutics is a major hurdle in developing nanoformulations [133]. The properties of nanoformulations are far different from that of their macro-analogs; thus, the current regulatory guidelines for drug development cannot be applied to nanoformulations. There is a need to update those guidelines keeping in mind the diverse and variable nature of each nanoformulation in terms of safety, efficacy, pharmacokinetics, pharmacodynamics, and toxicity [132]. Collectively, there are many more measures to be explored in terms of formulation, evaluation, application, and regulation of nanotherapeutics for stroke.

## 7. Conclusions and Future Prospects

Standard stroke treatment is not available, attributed to the characteristics of the drug delivery system such as poor BBB penetration, non-specific drug targeting, unfavorable ischemic tissue environment, and a smaller amount of the drug reaching the ischemic site. Nanotechnology provides scope to solve these issues and provide quality solutions for managing stroke. The characteristics of NPs, including their particle size and modifiable surface, allow them to be modified with agents that help cross the BBB. Moreover, generating nanotherapeutics that are responsive to a particular stimulus can be the future strategy for stroke management and diagnosis. For this, exploring the ischemic environment for distinct features like the presence of ROS, acidic pH, thrombus, and thrombus features like fibrin and active platelets can be used as internal stimuli for guiding the NPs toward the ischemic region and releasing drug payloads at the site. This is accomplished by using stimuli-responsive agents to prepare the NPs or ligating drugs with them. It ultimately increases the accumulation of NPs in the brain, thereby decreasing the exposure of NPs to the other tissues and reducing the toxicity (if any) and dose of administration. Stimuli-responsive targeting of NPs makes them safer towards the disease; hence, it is becoming a research hotspot. Although the proof of concept exists, still thorough research is required for these NPs to be in clinical practice. However, the development of novel stimuli-responsive NP-based therapeutics over the coming years will improve the outcomes in treating and diagnosing stroke.

## Figures and Tables

**Figure 1 pharmaceutics-15-01036-f001:**
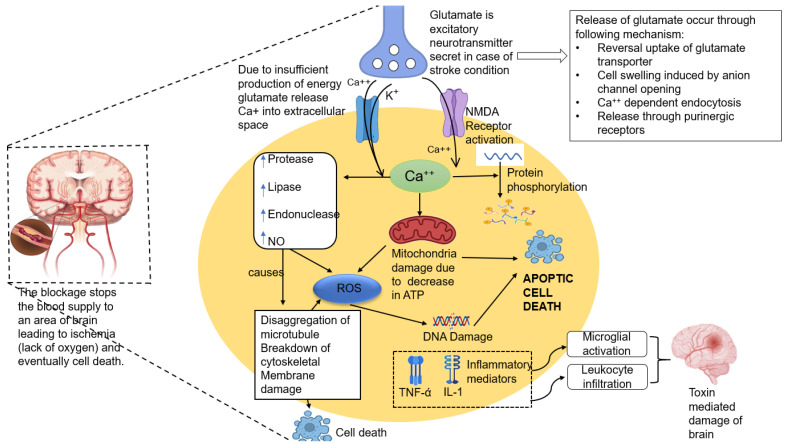
Pathophysiology of stroke at the cellular level.

**Figure 2 pharmaceutics-15-01036-f002:**
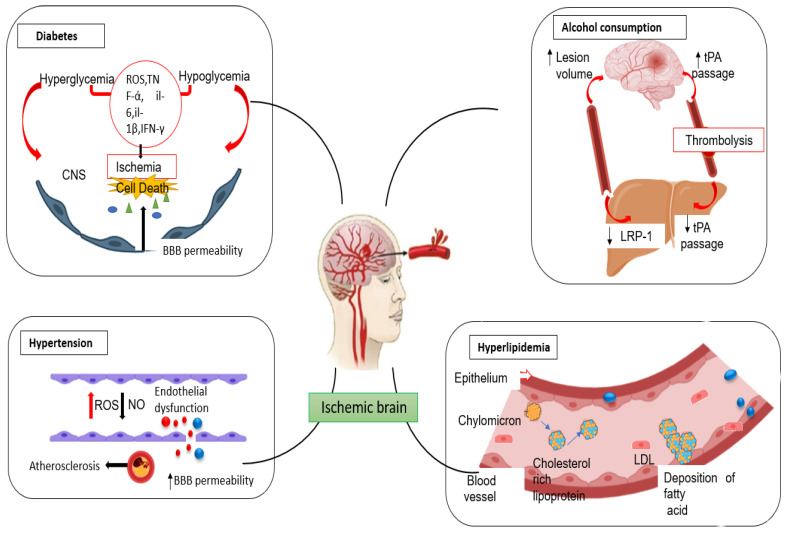
Factors associated with the development of stroke.

**Figure 3 pharmaceutics-15-01036-f003:**
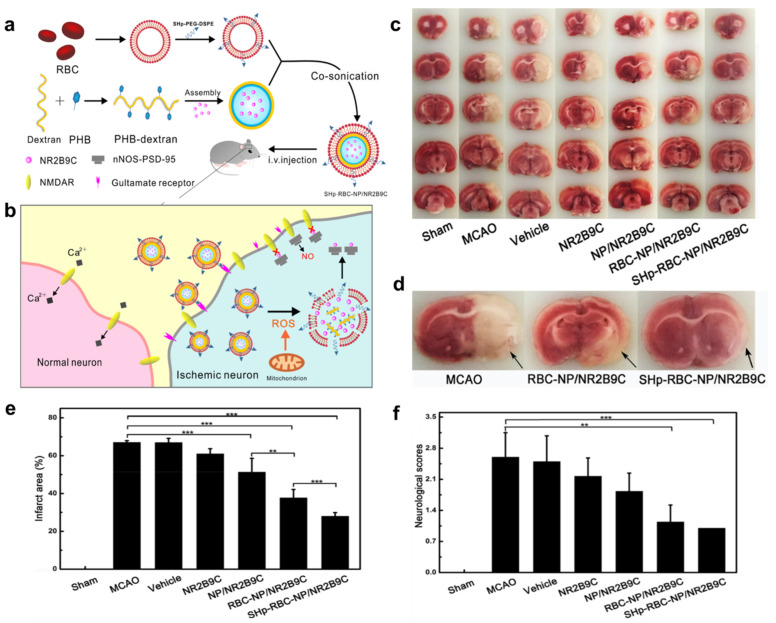
(**a**) Schematic design of the SHp-RBC-NP/NR2B9C, (**b**) ROS-responsive release of NR2B9C from SHp-RBC-NP/NR2B9C, (**c**) TTC brain section of Sham-operated group, MCAO, Vehicles, NR2B9C, NP/NR2B9C, RBC-NP/NR2B9C, and SHp-RBC-NP/NR2B9C. (**d**) Representative tissue slice, (**e**) brain infarct volume after 24 h, ** *p* < 0.01, *** *p* < 0.001 and (**f**) neuroscores of rats after cerebral ischemia, ** *p* < 0.01, *** *p* < 0.001 (adapted with permission from reference [81]).

**Figure 4 pharmaceutics-15-01036-f004:**
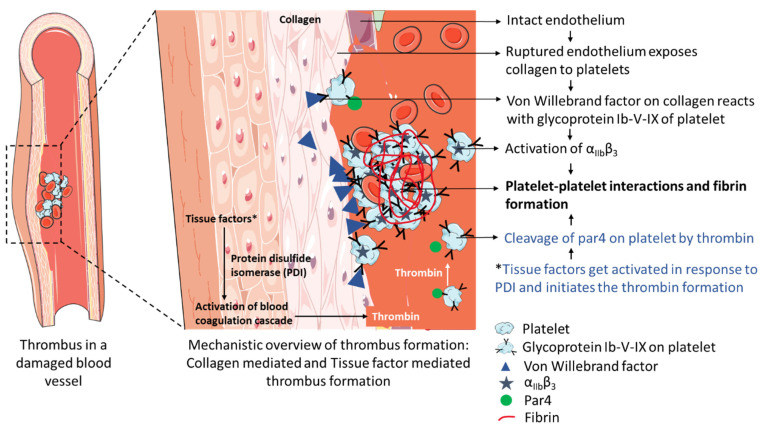
Thrombus formation in the blood vessel leads to the development of stroke.

**Figure 5 pharmaceutics-15-01036-f005:**
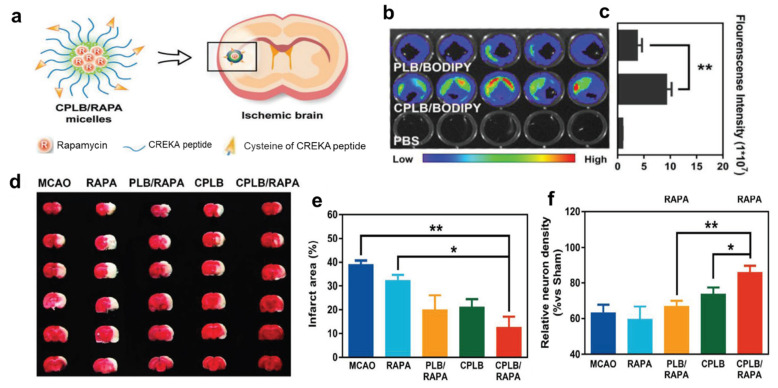
(**a**) Illustration of CPLB/RAPA micelle formation and modulation of the neurovascular unit in the ischemic brain. (**b**) The thrombus-targeting ability of CPLB/BODIPY micelles: the image of micelle binding with in vitro clots. (**c**) Quantification of the fluorescence intensity by measuring region of interest (ROI) in (**b**) (data presented as mean *n* = 5), ** *p* < 0.01 (**d**) Representative TTC-staining images of brain slices treated with different formulations, (**e**) Quantification of the infarct area in (**d**) (data presented as the percentage of white infarct area, *n* = 3), * *p* < 0.05, ** *p* < 0.01 (**f**) Quantification of the relative neuron density in ischemic penumbra by measuring the immunofluorescence intensity of NeuN (data presented as the ratio versus Sham group, *n* = 5), * *p* < 0.05, ** *p* < 0.01 (adapted with permission from reference [87]).

**Figure 6 pharmaceutics-15-01036-f006:**
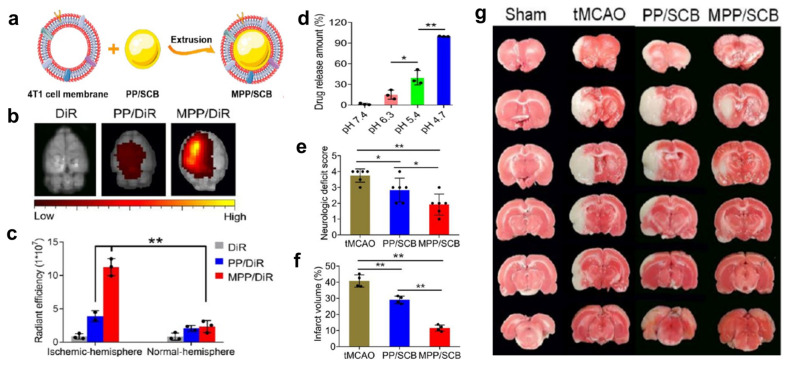
In vivo distribution and therapeutic evaluations of MPP/SCB NPs in tMCAO models, (**a**) MPP/SCB NPs fabrication, (**b**) ex vivo imaging of DiR, PP/DiR, and MPP/DiR in ischemia-affected brain tissues, (**c**) quantification of MPP distribution in the ischemic and normal hemisphere, ** *p* < 0.01, (**d**) in vitro SCB release from MPP/SCB NPs in citrate buffer with a different pH at 15 min, * *p* < 0.05, ** *p* < 0.01, (**e**) neurological deficit scores of the tMCAO rats from MPP/SCB treatment (*n* = 6), * *p* < 0.05, ** *p* < 0.01, (**f**) cerebral infarct volume in the sham group and tMCAO rats from each treatment (*n* = 4), ** *p* < 0.01, and (**g**) TTC staining images of the brain slices in sham-operated and tMCAO rats (adapted with permission from reference [91]).

**Figure 7 pharmaceutics-15-01036-f007:**
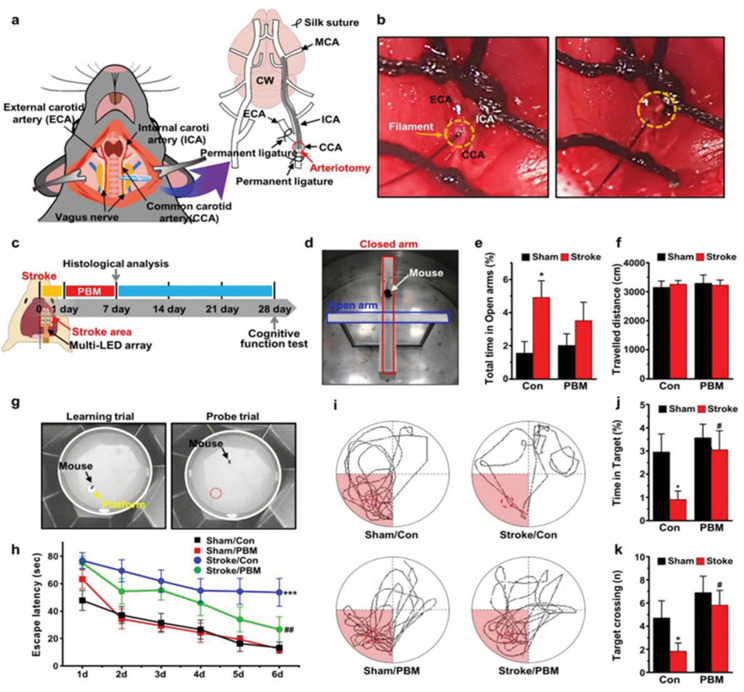
Impact of multi-LED array brain PBM at 630 nm on cognitive impairment after ischemic brain injury. (**a**) An illustration of the mouse model for MCAO/reperfusion. (**b**) Microscopic images 7-0 monofilament insertion coated with silicon from CCA to the ICA that covers the MCA. (**c**) Diagrammatic representation of an experimental PBM method in PSCI model utilizing a multi-LED array. (**d**) The raised plus maze test is used to quantify anxiety levels. (**e**) Duration spent in open arms; (**f**) total distance covered in the elevated plus maze throughout the 10-min period, * *p* < 0.05. (**g**) The Morris water maze test measures spatial learning and memory. (**h**) Average latency to locate the visible platform on day one and the concealed platform in the targeted quadrant on days 2–6 as a learning trial. *** *p* < 0.001, ## *p* < 0.01. (**i**) Typical swimming patterns observed in the probe trial period. (**j**) The amount of time spent in the desired quadrant, * *p* < 0.05 versus sham/control group; # *p* < 0.05 versus stroke/control group and (**k**) the desired number of crossings over the area where the concealed platform was first located, * *p* < 0.05 versus sham/control group; # *p* < 0.05 versus stroke/control group (Adapted with permission from reference [121]).

**Figure 8 pharmaceutics-15-01036-f008:**
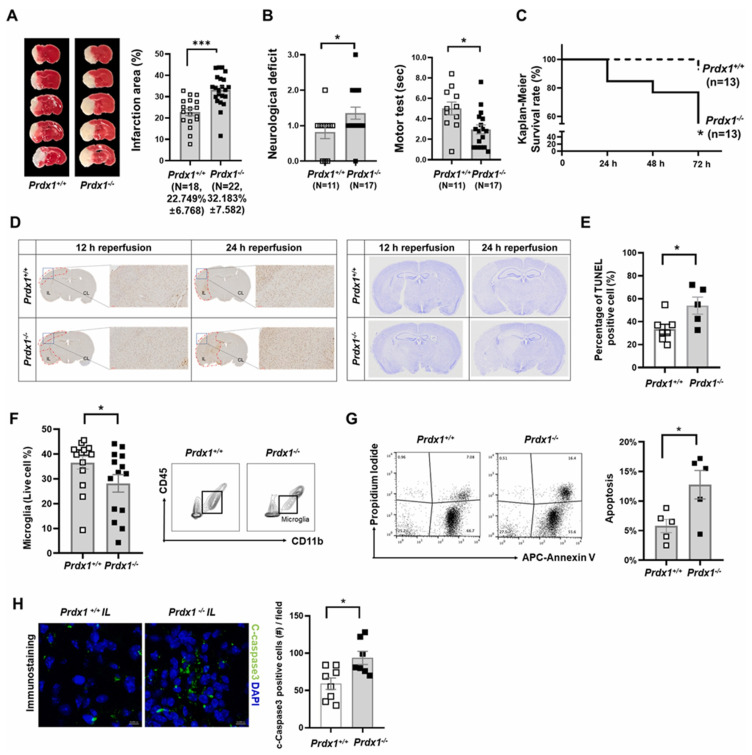
A deficit in Peroxiredoxin1 (Prdx1) enhances brain damage following the onset of a stroke. (**A**) Typical pictures of Prdx1+/+ (*n* = 18) and Prdx1/(*n* = 22) mouse brain slices stained with TTC, **** p* < 0.001. (**B**) Neurologic ratings for Prdx1+/+ (*n* = 11) and Prdx1/(*n* = 17) mice (**left panel**). Motor test results for Prdx1+/+ (*n* = 11) and Prdx1/(*n* = 17) mice, measured in seconds, * *p* < 0.05. (**C**) Kaplan–Meier Rates of survival (Gehan-Breslow-Wilcoxon) Prdx1+/+ (*n* = 13) and Prdx1/(*n* = 13) animals were tested at 24, 48, and 72 h following reperfusion. (**D**) Crystal violet staining (**right panel**) and TUNEL assay (**left panel**) of ischemic brain 12 and 24 h after stroke. (**E**) Bar graphs illustrating the TUNEL-positive cell percentage in Prdx1+/+ and Prdx1/+ brains of mice, * *p* < 0.05. (**F**) FACS examination of the fraction of microglia in the entire live cells of Prdx1+/+ (*n* = 13) and Prdx1/(*n* = 14) IL hemispheres, * *p* < 0.05. (**G**) FACS examination demonstrating Annexin V/PI staining in Prdx1+/+ (*n* = 5) and Prdx1/(*n* = 5) ipsilateral microglia, * *p* < 0.05. (**H**) Typical immunofluorescence pictures of c-caspase3 staining in Prdx1+/+ and Prdx1/mice infarction areas. Mice were used 24 h after ischemia, * *p* < 0.05 (Adapted with permission from reference [126]).

**Table 1 pharmaceutics-15-01036-t001:** Current treatment modalities in ischemic and hemorrhagic stroke.

Type of Stroke	Treatment	Therapeutic Agent	Mechanism of Action	Consequences	Complications	Ref.
Ischemic stroke	Thrombolysis	rtPA (Alteplase)	Activation of plasminogen to plasmin	Easy administration and fast reperfusion of coronary artery	May cause hemorrhagic stroke; to be administered within 3–4 h of stroke onset,	[30,31]
	Anticoagulants	Heparin	Deactivate thrombin and inhibit the clotting factors	Reduced recurrent stroke, deep vein thrombosis, and pulmonary embolism	May cause hemorrhage	[32,33]
	Anticoagulants	Warfarin	Vitamin K antagonist reduces clotting factors	Reduced recurrent stroke, deep vein thrombosis and pulmonary embolism	May cause hemorrhage	[33,34,35]
	Anticoagulants	Argatroban	Directly Inhibits the thrombin	Reduced recurrent stroke, deep vein thrombosis, and pulmonary embolism	May cause hemorrhage	[33,36,37]
	Antiplatelet therapy	Aspirin	Inhibits COX-1 involved in platelet adhesion	Secondary stroke prevention	Careful dose selection as higher doses may cause hemorrhage	[2,38,39]
	Antiplatelet therapy	Clopidogrel	P2Y12 receptor antagonist	Stroke management in aspirin-resistant/allergic patients	High risk for diabetics and coronary bypass surgery patients	[39,40,41]
	Anti-hyperlipidemics	Filgrastim and leucostim	Reduce cholesterol deposited in blood vessels	Neuroprotective, neuro-operative, and anti-inflammatory effects	More clear evidence is required	[42,43]
	Antioxidant	Edavarone	Free radical scavenger	Inhibit lipid peroxidation and lipoxygenase pathways	More clear evidence is required	[44,45]
	Anti-hypertensives	Nimodipine	Calcium channel blocker	Vasodilation and prevention of further damage	No significant prevention of cognitive decline	[46,47,48]
		Caffeinol	Inhibit GABA and NMDA receptors	Decline in volumes of cortical infarction	Daily administration causes caffeinol resistance	[49,50]
		Memantine	NMDA receptor antagonist	Reduces neuronal excitotoxicity	Careful dose selection as higher doses worsen the ischemic injury	[51,52]
	Endovascular thrombectomy	-	Surgical removal of thrombus	Patients not eligible for tPA treatment	Hemorrhage, vascular injury and vasospasms	[53,54,55]
Hemorrhagic stroke	Anti-hypertensives	Labetalol, Nicardipine, Hydralazine, Sodium nitroprusside, Esmolol, Enalaprilat, Nitroglycerin	Lowering the acute elevated blood pressure	Prevents hemorrhage	Risk of mortality, major disability, recurrent stroke, and cardiovascular events.	[56,57,58]
	Hemostatic therapy	Recombinant factor VIIa	Enhance coagulation and stop the bleeding	Most beneficial in the warfarin-related ICH	Effective for small hemorrhages	[59,60]
	Surgical evacuation	-	Surgical removal of blood and edema	Patients with hemorrhage in	Invasive procedure	[61]

**Table 2 pharmaceutics-15-01036-t002:** Stimuli-responsive nanotherapeutics for diagnosis and treatment of stroke.

Stimuli	Detail of Stimuli	Nanomaterial	Stimuli-Responsive Moiety	In Vivo Animal Model	Results	Ref.
Oxidative stress	H_2_O_2_	SHp-RBC-NP/NR2B9C NPs	PHB-Dextran	MCAO rat model in male Sprague Dawley rats	Significant reduction in cerebral infarct volume	[81]
	ROS	PEG-PPS-NPs	PPS	MCAO mouse model	Reduced neuronal loss post stroke	[82]
	H_2_O_2_	PVAX NPs	Copolyoxalate	Mouse ischemic reperfusion injury model	Reduced levels of TNF-α, IL-1β, suggesting anti-inflammatory activity in ischemic stroke	[83]
	H_2_O_2_	HPOX-Rubrene NPs	Copolyoxalate	Mouse ischemic reperfusion injury model	Increased luminescence at ischemic site	[84]
	HOCl and ˙OH	IR-LnNPs	IR-783	LPS-induced oxidative stress model in mice	Decreased luminescence at ischemic site	[85]
Thrombosis	Fibrin	SOD1 *cl*-nanozymes	*cl*-nanozymes	MCAO rat model in male Sprague Dawley rats	High SOD1 reaching to thrombus site and scavenging the ROS	[86]
	Fibrin	CPLB/RAPA NPs	C, i.e., CREKA peptide	transient-MCAO model in male Sprague Dawley rats	Higher rapamycin release and ROS scavenging at thrombotic site	[87]
	Thrombin	Glyburide loaded AMD3100-PEG-PCL-T-PEG NPs	NH2-norleucine-TPRSFL-CSH (T)	MCAO model in Male C57BL/6 mice	Significant reduction in cerebral infarct volume and improved neurological score	[88]
	Fibrin	EP-2104R NPs	EP-2104R	Fibrin from pig, dog, rabbit, rat, or mouse plasma	MRI contrast agent for the detection of a blood clot in stroke	[89]
Acidic pH	pH 6.9–7.1	Chitosan-nimodipine NPs	Chitosan	MCAO rat model in male Sprague Dawley rats	Cerebral vasodilation and increasing the CBF	[90]
	pH 4.7 and 5.4	MPP/SCB NPs	PEG-PDPA polymer	transient-MCAO model in male Sprague Dawley rats	NPs concentrated in ischemic brain release SCB and reduce infarct volume to 69.9%	[91]
	pH < 7.0	PEG-coated 4-amino-TEMPO NPs	4-amino- group on TEMPO coblock polymer	MCAO rat models in male Sprague Dawley rats	Significant reduction in cerebral infarct volume, the decline in ROS, hence less protein oxidation	[92]
	pH ≤ 6.8	PEG-PAE-Fe_3_O_4_ NPs	PAE	MCAO rat models in male Sprague Dawley rats	Increased MRI signals from ischemic tissue	[93]
	pH ≤ 6.8	methyl-PEG-PAEA-Fe_3_O_4_ NPs	PAEA	MCAO rat models in male Sprague Dawley rats	Increased MRI signals from ischemic tissue	[94]
Light	NIR	Nd3+-doped upconversion nanoparticles	Neodymium	MCAO model in mice	Increased angiogenesis, decreased infraction, better recovery in neuronal function	[95]
	NIR	gold@mesoporous silica core–shell nanospheres	Gold	Carrageenan-induced tail thrombus model	Enhanced thrombolysis, reduced side effects with high effectiveness	[96]
Enzyme	-	-	cyclic GMP-AMP (cGAMP) synthase (cGAS)	MCAO mice model in C57BL/6 mice	cGAS inhibition reduces brain inflammation	[97]
	-	Platelet microparticle (PMP)-inspired nanovesicles	Phospholipase-A_2_	FeCl3-induced carotid thrombosis model	Effective targeted thrombolysis with site-specific therapy	[98]
Ultrasound	Ultrasonic waves	-	low-intensity transcranial ultrasound stimulation (LITUS)	MCAO rat models in Sprague Dawley rats	Inhibition of apparent diffusion coefficient decrease with LITUS	[99]
	Ultrasonic waves	MMB-SiO_2_-tPA nanoparticles	Microbubbles	FeCl3-induced femoral vein thrombosis model	Enhanced lytic rate with better therapeutic effect	[100]

## Data Availability

Not applicable.

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
