# Peer review of "Stimuli-Responsive Nanotherapeutics for Treatment and Diagnosis of Stroke"

_pharmaceutics, 2023, doi:10.3390/pharmaceutics15041036_

Round 1

Reviewer 1 Report

This article represents a comprehensive and very well written review of the use of various types of stimuli-responsive nanoparticles in the treatment and diagnosis of stroke. The review is well structured, involves all types of stimuli I am aware about, and demonstrates literature findings with details. I believe the review is in a very good status to be proceeded with.

Author Response

Authors response attached

Reviewer 2 Report

As I reviewed this work “Stimuli-responsive nanotherapeutics for treatment and diagnosis of stroke”, the authors worked on the important topic of “Stroke” as the second most common medical emergency and constitutes a significant cause of global morbidity. The described the deficiency of conventional stroke treatment strategies to survive the patients. Then, they claimed that employing the nanoparticles toward the ischemic tissues by making them stimuli-responsive can be a turning point in managing stroke. Details of this work are informative and the authors declared what they wanted for following their purpose. I can recommend this work for publishing, but I may still advice the authors to make some small corrections to their work.

In the part of nano-therapeutic, definitions are not clear. You may first refer to the advantages of nanoparticles in the sensing and drug delivery related functions, then you can move to explaining their single-standing functions for making such important responses. I can introduce you the following works, doi:   https://doi.org/10.22034/chemm.2021.121496  , doi:  https://doi.org/10.1016/j.diamond.2023.109749  and doi:  https://doi.org/10.1016/j.comptc.2022.113866  , you may include in your work to support the first descriptions of nanoparticles for working in the drug-related systems.

Author Response

Authors response attached

Reviewer 3 Report

In the review article presented for review, Tekade et al. described the phenomenon of stroke, its causes, limitations resulting from the currently known methods of its therapy, as well as the use of nanotherapeutics in the diagnosis and treatment of stroke effects. I consider the entire study to be very well prepared in terms of content and editorial. I strongly support publication of this manuscript in Pharmaceutics without revisions.

Author Response

Authors response attached
